# Catalytic Production of Glycolic Acid from Glycerol Oxidation: An Optimization Using Response Surface Methodology [†]

Claudia Patricia Tavera Ruiz [1,2] , Franck Dumeignil [1] and Mickaël Capron [1,*]

1    Univ. Lille, CRNS, Centrale Lille, ENSCL, Univ. Artois, UMR 8181-UCCS-Unité de Catalyse et Chimie du Solide, F-59000 Lille, France; ctavera3@udi.edu.co (C.P.T.R.); franck.dumeignil@univ-lille.fr (F.D.)

2    FIELDS, Universidad de Investigación y Desarrollo, Calle 9 # 23-55, Bucaramanga, Colombia

*    Correspondence: mickael.capron@univ-lille.fr

†    In memory of Prof. Elzbieta Skrzynska.

**Abstract:** This study aimed at optimizing the production of glycolic acid from glycerol catalytic oxidation over a silver catalyst supported on a mixed cerium-zirconium oxide, to progress towards the industrialization of a derived process. Optimization of the operating conditions was performed using the response surface methodology. We concluded that the production of glycolic acid depends mainly of glycerol concentration, NaOH/glycerol ratio, catalyst/glycerol ratio, and $O_2$/glycerol ratio. The optimal conditions we found were a temperature of 60 °C, a NaOH/glycerol molar ratio of 2, an $O_2$/glycerol molar ratio of 0.23, and a catalyst/glycerol mass ratio of 0.07. With these optimal conditions, it was possible to increase the glycerol concentration from 0.3 M to 2 M, obtaining an increase in the concentration of glycolic acid in the liquid fraction, from 0.27 mol/L of glycolic acid (with initial glycerol solution 0.3 M) to 0.88 mol/L (with initial solution 2 M), while keeping a 100% glycerol conversion.

**Keywords:** glycerol valorization; optimization process; glycolic acid production

## 1. Introduction

Due to the oil crisis and the negative environmental impacts that the production and use of fossil fuels represent, the consumption of renewable energies has become a priority worldwide and research towards that direction has exhibited rapid growth and evolution. In the last decade, biodiesel has become an important alternative to fossil fuels, which has led to an increase in its production from different vegetable and animal fats. This high production of biodiesel leads to an excess production of its main by-product: crude glycerol (10 kg of biodiesel generates ca. 1 kg of glycerol) [1–3]. The excess of crude glycerol has caused a fall in the sale price (less than 0.5 €/kg for the lowest quality), as well as environmental and economic issues due to the fact that its final disposal is quite difficult and expensive [1,4].

Different glycerol valorization pathways have been studied to generate alternative energy carriers and to obtain high-value-added compounds of interest in different industrial applications, such as food, cosmetics, paints, and pharmaceuticals, among others [1,5].

The various identified routes to generate energy are combustion, pyrolysis, and hydrogen production (through reforming and/or partial oxidation). Combustion is an inexpensive process and is carried out without a catalyst, but glycerol presents a low heating value (18 MJ/kg) [5] and the production of $CO_2$ is an environmental disadvantage. On the other hand, pyrolysis requires catalysts, from which $H_2$/syngas, aldehydes, ketones, and aromatics of low molecular weight can be obtained. However, $H_2$ yields are very low, and optimization of the yields is required. Finally, different routes have been studied to obtain hydrogen, such as steam reforming, partial oxidation, autothermal reforming, aqueous-phase reforming, and supercritical water reforming [6]; such routes are considered as a promising for glycerol valorization.

For the production of molecules with high added value for industrial applications, different reactions have been studied, among which selective oxidation, etherification, hydrogenolysis, dehydration, and fermentation, notably to *1,3*-propanediol, stand out [7–12].

Within all these glycerol valorization routes, oxidation in the liquid phase using metal catalysts has been studied extensively, targeting various products of industrial value (i.e., glyceric acid (GA), glycolic acid (GLYA), or tartronic acid (TA)) [3,13–21], among which GLYA has the greatest commercial value and the highest price (ca. 570 €/kg) [22]. It is used in a wide range of applications in the cosmetics, pharmaceutical, and food industries [1,4]. For example, GLYA is currently widely used in the cosmetic industry as an active ingredient in different high-demand products, such as a peeling agents, anti-aging creams and acne removal creams, photoaging, melasma, and other skin-related diseases [23].

Glycerol oxidation in the liquid phase using heterogeneous catalysis is a promising process for transforming glycerol into high-value-added products. Previous studies have shown that using silver-based catalysts, it is possible to obtain a high glycerol conversion and also good selectivity towards GLYA [3,13,21]. In these previous investigations, a 0.3 M glycerol-water solution was used, and NaOH was added to promote the reaction. Even if the results obtained with these studies have so far been successful and promising, it is necessary before envisioning industrialization to solve some technical points, such as increasing the initial concentration of glycerol, decreasing the amount of water to facilitate the separation of the final products, reducing the amount of soda in the solution, and maximizing the amount of GLYA.

The present study aimed at finding conditions that allow lifting these technical locks, thus progressing toward industrialization of the process. For this, a three-stage strategy was used: First, we identified the most sensitive reaction parameters for GLYA production maximization using the response surface methodology, which was followed by an optimization of the as-identified reaction parameters. Finally, the optimal conditions we found were tested with increased glycerol concentrations.

## 2. Results and Discussion

### 2.1. Catalyst Characterization

The diffractograms of the $Ce_{0.75}Zr_{0.25}O_2$ support and the 5% $Ag/Ce_{0.75}Zr_{0.25}O_2$ catalyst are shown in Figure 1. The characteristic peaks of the cubic $Ce_{0.75}Zr_{0.25}O_2$ phase were detected in both cases. The highest peak was observed at $2\Theta = 28.66°$, which corresponded to the (111) plane. We also identified the main characteristic peaks of the (200) at $2\Theta = 33.19°$, (220) at $2\Theta = 47.74°$, (311) at $2\Theta = 56.68°$, (222) at $2\Theta = 59.44°$, (400) at $2\Theta = 70.06°$, and (331) planes at $2\Theta = 76.99°$, which suggested the presence of a pure cubic phase (JCPDS file: 04-011-8939) [24]. The size of the support crystallites was estimated from the peak at $2\Theta = 28.6°$, obtaining a value of 5.8 nm. The characteristic peaks of the precursors were not found, which suggested a rather high purity of the obtained support. After silver deposition, the same peaks were still observed, indicating that the support structure was preserved upon deposition. A small peak located at $2\Theta = 38°$, corresponding to the (111) plane of the metallic silver phase, was then detected (JCPDS file: 01-089-3722). No peaks due to impurities or silver precursors were observed.

Figure 2 shows the nitrogen adsorption/desorption profiles and the pore volume distribution of the support before and after silver deposition. The obtained adsorption isotherms (Figure 2a,b) were type IV corresponding to mesoporous solids. It can also be seen that after silver deposition, the textural properties of the support were preserved.

Table 1 shows the specific surface area values, the pore diameter, and the pore volume determined by the BJH method for the support before and after silver loading. Upon silver loading, the specific surface area decreased slightly from 97 $m^2/g$ to 86 $m^2/g$. On the other hand, pore size and pore volume did not show significant change, meaning that silver was well dispersed on the surface of the support and did not cause blockage of the pores.

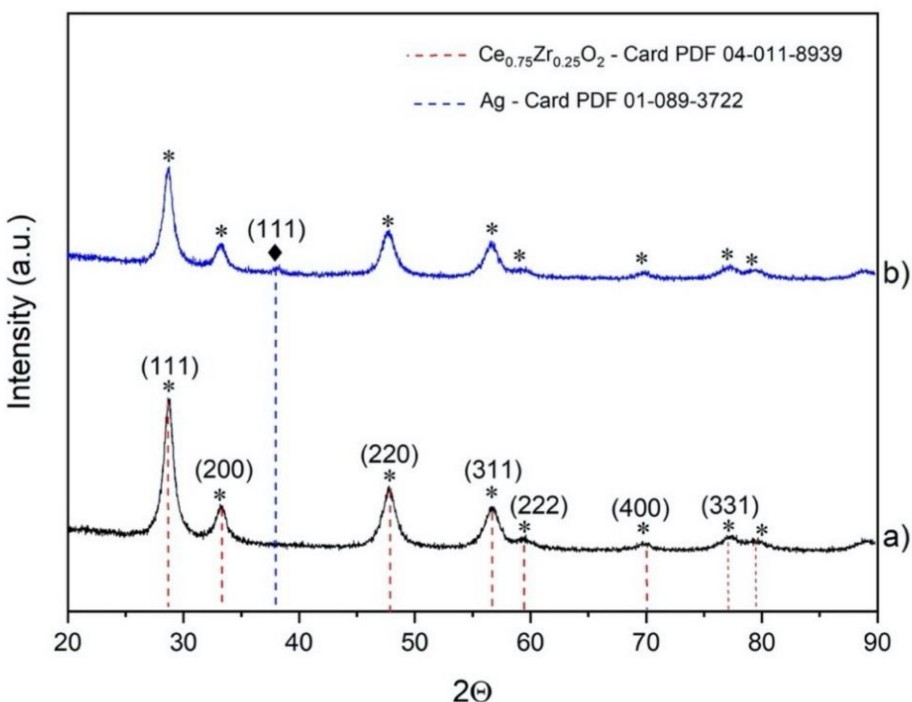

**Figure 1.** XRD pattern of (**a**) synthetized support $Ce_{0.75}Zr_{0.25}O_2$ and (**b**) synthetized catalysts: 5 wt% of Ag impregnated on $Ce_{0.75}Zr_{0.25}O_2$ catalyst. Symbols correspond to: (∗) Main peaks of the $Ce_{0.75}Zr_{0.25}O_2$ structure; and main peak of silver particles (◆).

**Table 1.** Textural properties for the support before and after silver loading.

| Catalyst | Specific Surface Area (m$^2$/g) | Pore Size (nm) | Pore Volume (cm$^3$/g) |
|---|---|---|---|
| $Ce_{0.75}Zr_{0.25}O_2$ support | 97 | 11.2 | 0.27 |
| 5% $Ag/Ce_{0.75}Zr_{0.25}O_2$ | 86 | 11.7 | 0.25 |

The XPS spectra for cerium (Ce 3d) are shown in Figure 3. The obtained peaks were decomposed in eight peaks corresponding to the spin–orbit doublets: 3d 5/2 and 3d 3/2 (Figure 3). The peaks v′, u′, v$^0$, and u$^0$ correspond to $Ce^{3+}$ species, while the peaks v, u, v‴, u‴, v″, and u‴ correspond to $Ce^{4+}$ species. Likewise, the proportion of $Ce^{3+}$ and $Ce^{4+}$ with respect to total cerium was calculated (Table 2), finding that for the $Ce_{0.75}Zr_{0.25}O_2$ support there was 83.2% of $Ce^{4+}$, and for 5%$Ag/Ce_{0.75}Zr_{0.25}O$ there was 82%. This may suggest oxygen vacancies or O atoms relaxations resulting from silver deposition [25,26].

**Table 2.** Composition parameters obtained from XPS analysis.

| Catalyst | Mass Conc. (wt.%) | | | | Atomic Concentration (%) | | | | |
|---|---|---|---|---|---|---|---|---|---|
| | % Ag | Ce$^{+3}$ | Ce$^{+4}$ | Total Ce | % Zr (Zr3d) | % O (O1s) | % Ag (Ag3d) | Atomic Ratio | |
| | | | | | | | | Ce$^{4+}$/Zr | Ce$^{3+}$/Zr |
| $Ce_{0.75}Zr_{0.25}O_2$ support | - | 3.75 | 18.75 | 22.39 | 5.63 | 72.03 | - | 3.33 | 0.67 |
| 5%$Ag/Ce_{0.75}Zr_{0.25}O_2$ | 4.64 | 4.22 | 18.43 | 22.65 | 5.47 | 68.44 | 3.60 | 3.37 | 0.77 |

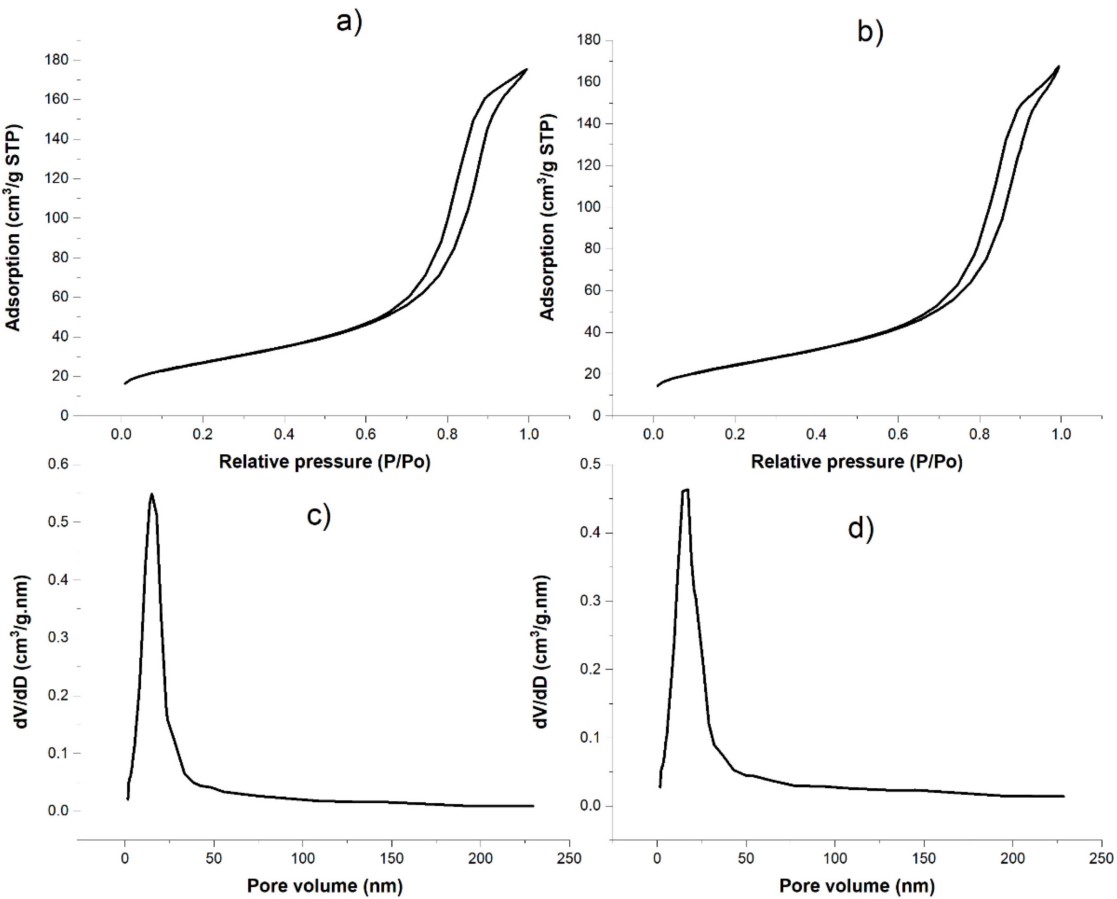

**Figure 2.** Nitrogen adsorption/desorption profiles for (**a**) $Ce_{0.75}Zr_{0.25}O_2$ support and (**b**) $5\%Ag/Ce_{0.75}Zr_{0.25}O_2$ catalyst, and pore volume distribution of (**c**) $Ce_{0.75}Zr_{0.25}O_2$ support and (**d**) $5\%Ag/Ce_{0.75}Zr_{0.25}O_2$.

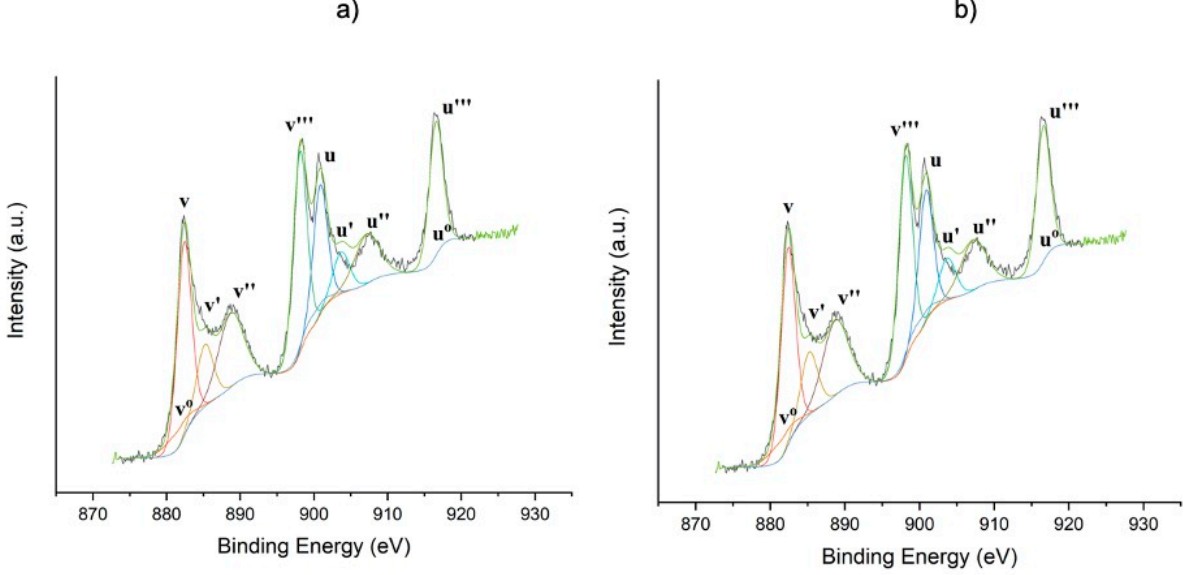

**Figure 3.** Ce 3d XPS spectra of (**a**) $Ce_{0.75}Zr_{0.25}O_2$ support and (**b**) $5\%Ag/Ce_{0.75}Zr_{0.25}O_2$ catalyst.

Figure 4 shows the XPS spectra of $Ce_{0.75}Zr_{0.25}O_2$ support and $5\%Ag/Ce_{0.75}Zr_{0.25}O_2$ catalyst for O1s, Zr 3d, and Ag 3d peaks. The obtained peak positions were similar to those reported in the literature. For silver (Ag 3d), the peaks were located at binding energies (BEs) equal to 368.1 eV (Ag 3d 5/2) and 374.1 eV (Ag 3d 3/2), which corresponded to metallic silver [13]. For zirconium (Zr 3d), the peaks were at Bes = 182 and 184.4 eV, corresponding to $ZrO_2$. No significant changes were observed on the zirconium peaks before and after silver deposition.

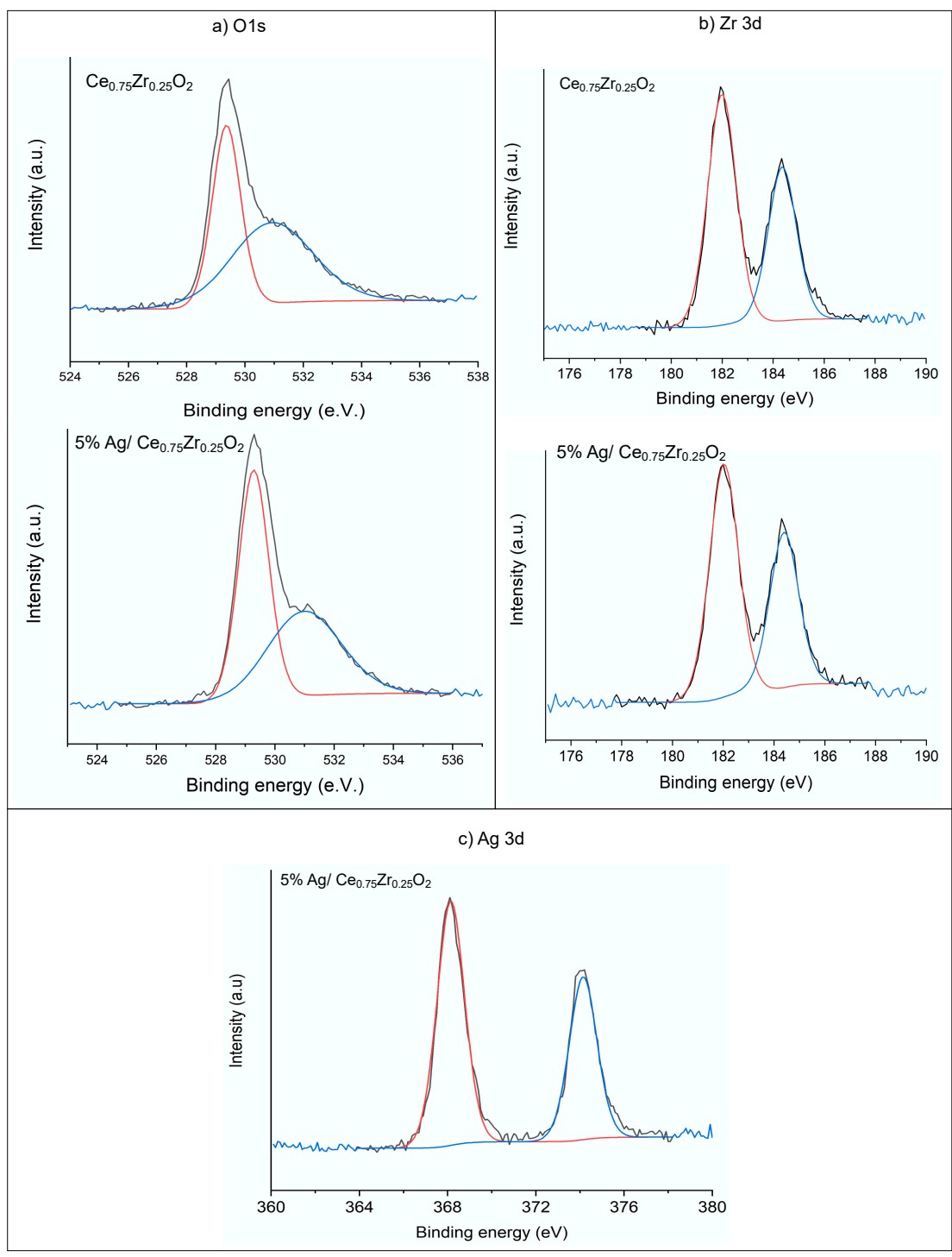

**Figure 4.** XPS spectra of $Ce_{0.75}Zr_{0.25}O_2$ support and $5\%Ag/Ce_{0.75}Zr_{0.25}O_2$ catalyst for (**a**) O 1s peaks, (**b**) Zr 3d peaks, and (**c**) Ag 3d peaks.

For the O1s region, no significant changes were found in the shape of the peaks upon silver deposition. Two peaks were evidenced at BE = 529 and 531 eV, which were characteristic of lattice and lattice-defective oxygen vacancies, respectively [27]. Regarding the atomic concentration of oxygen (Table 2), a slight decrease after silver deposition was observed, which supported the hypothesis of formation of oxygen vacancies.

Table 2 shows the atomic concentration (%) determined by XPS surface analysis, before and after silver deposition. The XPS analysis evidenced that the surface silver loading (wt.%) was a little bit lower (4.64 wt.%) than the theoretical one of 5 wt.%.

### 2.2. Experimental Tests and Optimization

A preliminary benchmark test was carried out with the initial conditions that corresponded to those evaluated in previous investigations carried out in our laboratory [5]. These conditions corresponded to a 0.3 M glycerol solution, a reaction temperature of 60 °C, 5 bar of oxygen, 0.5 g of catalyst, and a NaOH/glycerol molar ratio of 4. These tests were performed in triplicate to prove reproducibility of the system. The results obtained at these initial operating conditions are shown in Figure 5. The carbon balance was around 100%. After two hours of reaction, we observed almost full conversion (>98%) with a high selectivity to GLYA (69%). The GLYA yield (i.e., 68%) obtained with the 5%Ag/Ce$_{0.75}$Zr$_{0.25}$O$_2$ catalyst was the highest one reported in the literature so far, to the best of our knowledge. [3,13].

The other products formed in lower selectivity were GA (10%), which is also a high-added-value product [16,28], and formic acid (FA, 20%), which resulted from C–C bond cleavage during the GLYA production. A yield of 63.7% of GLYA was thus obtained under these conditions (bottom of Figure 5).

Experimental tests to optimize the glycolic acid were carried out in two stages. In the first stage, we determined the most sensitive experimental parameters. For this, a second-degree central composite design (CCD) was developed, which was elaborated using XLSTAT [29].

In the first CCD, named CCD 1, the studied parameters (variables) were glycerol concentration, NaOH/glycerol molar ratio, reaction temperature, and glycerol/catalyst ratio. The response variable was the yield of GLYA. In this experimental design, twenty-five observations were proposed. The levels of each variable are shown in Table 3. Such levels were defined according to previous results obtained by our laboratory and according to bibliographical data.

**Table 3.** Variables of study in the central composite design (CCD) 1.

| Variable | Minimum Level | Maximum Level |
|---|---|---|
| Glycerol Concentration | 0.3 M | 2.5 M |
| NaOH/glycerol molar ratio | 0 | 4 |
| Catalyst/glycerol ratio | 0.1 | 0.9 |
| Reaction temperature | 40 °C | 120 °C |

The experimental test results were analyzed by an ANOVA analysis, in which the most influential variables and their interactions were found.

In order to obtain the maximum yield of GLYA by increasing the glycerol concentration and minimizing the amount of NaOH, the experimental tests of the CCD 1 design were carried out for 300 min. It was found for all the tests that once the maximum conversion was reached, the production of glycolic acid remained stable until the end of the 300 min test, like it was observed in the preliminary benchmark test (Figure 5). Table 4 presents the operating conditions of each test, as well as the obtained yield of glycolic acid.

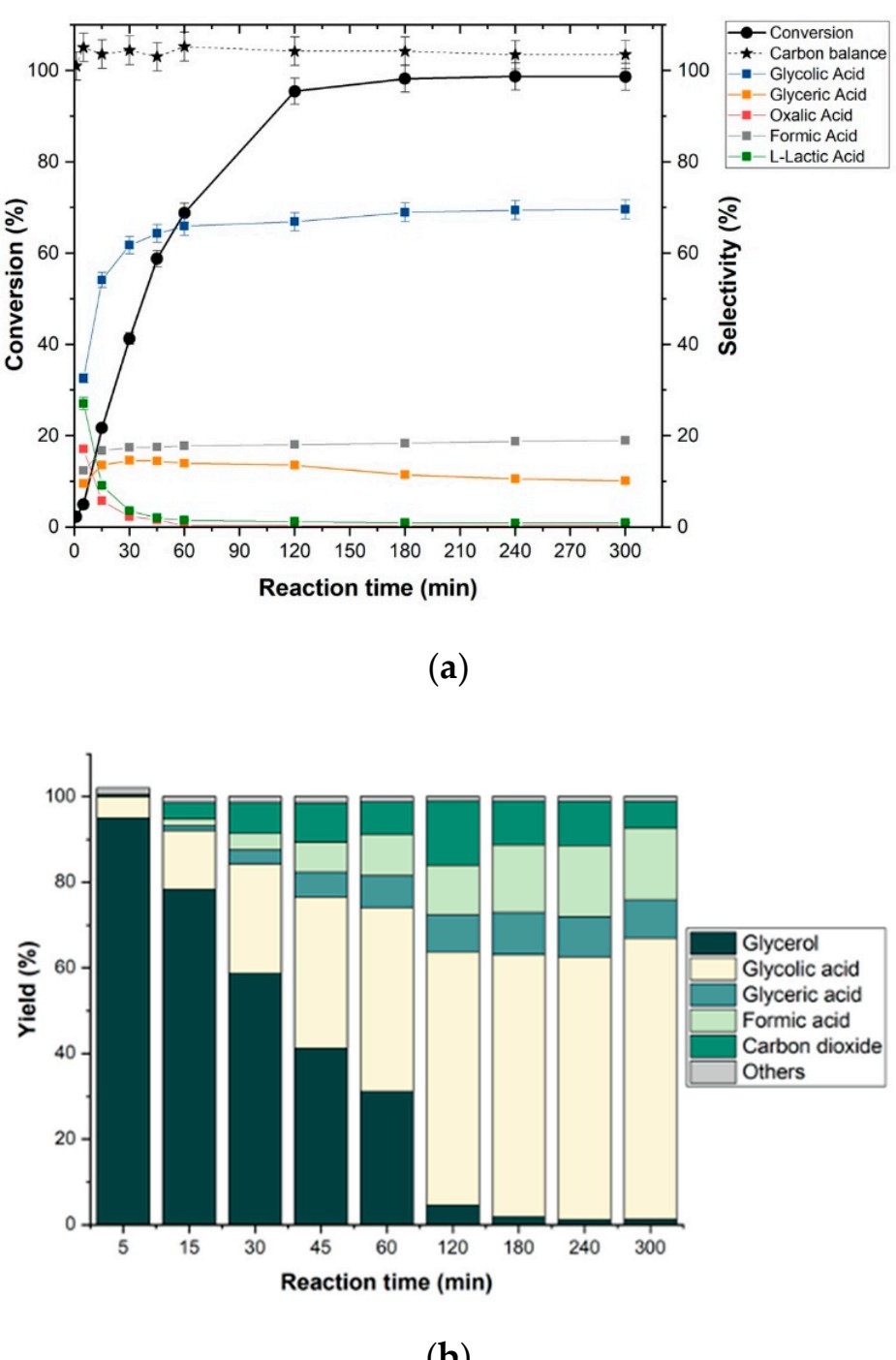

**Figure 5.** Experimental results obtained for benchmark test at previously determined operating conditions: 0.3 M glycerol solution, 60 °C, 5 bar, 0.5 g of catalyst, and a NaOH/glycerol molar ratio of 4 (**a**) glycerol conversion and products selectivity and (**b**) products yield.

**Table 4.** Experimental results obtained at each operating conditions of the CCD 1.

| Test | Glycerol Concentration (mol/L) | NaOH/Glycerol Molar Ratio (mol/mol) | Reaction Temperature (°C) | Catalyst/Glycerol Ratio | Glycolic Acid Yield (%) |
|---|---|---|---|---|---|
| 1 | 1 | 1 | 60 | 0.3 | 30.95 |
| 2 | 2 | 1 | 60 | 0.3 | 14.10 |
| 3 | 1 | 3 | 60 | 0.3 | 16.96 |
| 4 | 2 | 3 | 60 | 0.3 | 6.32 |
| 5 | 1 | 1 | 100 | 0.3 | 26.23 |
| 6 | 2 | 1 | 100 | 0.3 | 19.70 |
| 7 | 1 | 3 | 100 | 0.3 | 28.41 |
| 8 | 2 | 3 | 100 | 0.3 | 19.32 |
| 9 | 1 | 1 | 60 | 0.7 | 28.27 |
| 10 | 2 | 1 | 60 | 0.7 | 23.70 |
| 11 | 1 | 3 | 60 | 0.7 | 37.14 |
| 12 | 2 | 3 | 60 | 0.7 | 11.26 |
| 13 | 1 | 1 | 100 | 0.7 | 25.8 |
| 14 | 2 | 1 | 100 | 0.7 | 8.02 |
| 15 | 1 | 3 | 100 | 0.7 | 18.36 |
| 16 | 2 | 3 | 100 | 0.7 | 12.06 |
| 17 | 0.5 | 2 | 80 | 0.5 | 52.93 |
| 18 | 2.5 | 2 | 80 | 0.5 | 9.48 |
| 19 | 1.5 | 0 | 80 | 0.5 | 0.03 |
| 20 | 1.5 | 4 | 80 | 0.5 | 9.75 |
| 21 | 1.5 | 2 | 40 | 0.5 | 12.45 |
| 22 | 1.5 | 2 | 120 | 0.5 | 28.51 |
| 23 | 1.5 | 2 | 80 | 0.1 | 23.04 |
| 24 | 1.5 | 2 | 80 | 0.9 | 36.16 |
| 25 | 1.5 | 2 | 80 | 0.5 | 24.48 |

Upon calculation of the test statistics, the *p*-value was obtained, which is defined as the probability of observing the given value of the test statistic, being a measure of evidence against the null hypothesis [30,31]. The value to reject the null hypothesis was 0.05, which means that no significant difference existed. Table 5 shows the results of the statistical analysis of variance (ANOVA) for GLYA with four variables and their interactions. In this case, two effects had *p*-values less than 0.05, indicating that they were significantly different from zero at the 95.0% confidence level.

**Table 5.** Analysis of variance for glycolic acid (GLYA) yield.

| Source | Sum of Squares | Df | Mean Square | F-Ratio | *p*-Value |
|---|---|---|---|---|---|
| A: Glycerol concentration | 1515.28 | 1 | 1515.28 | 27.81 | 0.0003 |
| B: NaOH/glycerol | 5.3053 | 1 | 5.3053 | 0.10 | 0.7609 |
| C: Temperature | 13.3846 | 1 | 13.3846 | 0.25 | 0.6299 |
| D: Catalyst qty | 27.2302 | 1 | 27.2302 | 0.50 | 0.4943 |
| AA | 26.4785 | 1 | 26.4785 | 0.49 | 0.5002 |
| AB | 0.421625 | 1 | 0.421625 | 0.01 | 0.9315 |
| AC | 31.1348 | 1 | 31.1348 | 0.57 | 0.4656 |
| AD | 3.94409 | 1 | 3.94409 | 0.07 | 0.7929 |
| BB | 288.044 | 1 | 288.044 | 5.29 | 0.0421 |
| BC | 48.7598 | 1 | 48.7598 | 0.89 | 0.3645 |
| BD | 18.0048 | 1 | 18.0048 | 0.33 | 0.5770 |
| CC | 14.9658 | 1 | 14.9658 | 0.27 | 0.6106 |
| CD | 217.575 | 1 | 217.575 | 3.99 | 0.0710 |
| DD | 14.4152 | 1 | 14.4152 | 0.26 | 0.6172 |
| Total error | 599.423 | 11 | 54.493 | | |
| Total (corr.) | 3349.57 | 25 | | | |
| *R*-Squared | | | 82.11 | | |

The *R*-squared statistic indicated that the model as fitted explains ca. 82% of the variability in GLYA yield. Since the *p*-value was larger than 5.0%, there was no indication of serial autocorrelation in the residuals at the 5.0% significance level. Those variables with a *p*-value less than 0.05 indicated a significant correlation, being the variables with the largest influence on the yield of GLYA. The most influential variables were glycerol concentration and the interaction (NaOH/Glycerol)$^2$, with *p*-values of 0.0003 and 0.0421, respectively.

Figure 6 shows the combination of factor levels, which maximize the desirable function (GLYA yield) over the indicated region. It also shows the combination of the factors' values at which that optimum was achieved. With these results, we could find the optimal influential variables and the corresponding GLYA yield. The optimal as-determined values were 0.51 M for glycerol concentration, 61.4 °C for the temperature, a NaOH/glycerol ratio equal to 2.22, and 0.73 g of catalyst. With these results, compared to the benchmark, we could then decrease the NaOH/glycerol ratio from 4 to 2 and increase the glycerol concentration from 0.3 M to 0.5 M, which corresponded to an overall increase of glycolic acid recovery from 0.27 mol/L g to 0.44 mol/L in the same reaction volume (200 mL).

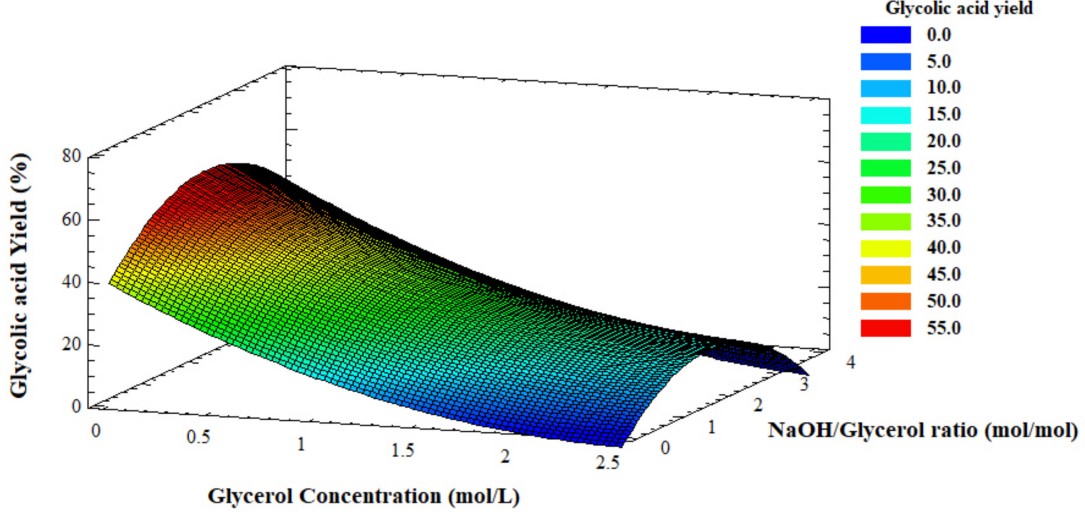

**Figure 6.** Surface response for GLYA yield evaluating the most influential variables (i.e., glycerol concentration and NaOH/glycerol ratio).

Once the optimal conditions were obtained, we carried out tests increasing the glycerol concentration to 1.5 M and 2.5 M at the optimal conditions obtained in the previous study—temperature (60 °C) and NaOH/Glycerol ratio (2), $O_2$ pressure of 5 bar, and a quantity of catalyst equal to 0.7 g. The obtained results are shown in Figure 7. It was found that when the concentration of glycerol was increased, the conversion decreased considerably, leading to a lower GLYA yield. In addition, it could be found that at higher glycerol concentrations, lactic acid (LA) was detected, which was produced due to an oxygen deficiency in the medium (Figure 8). The optimal conditions obtained in the previous optimization could not be directly transposed to higher glycerol concentrations.

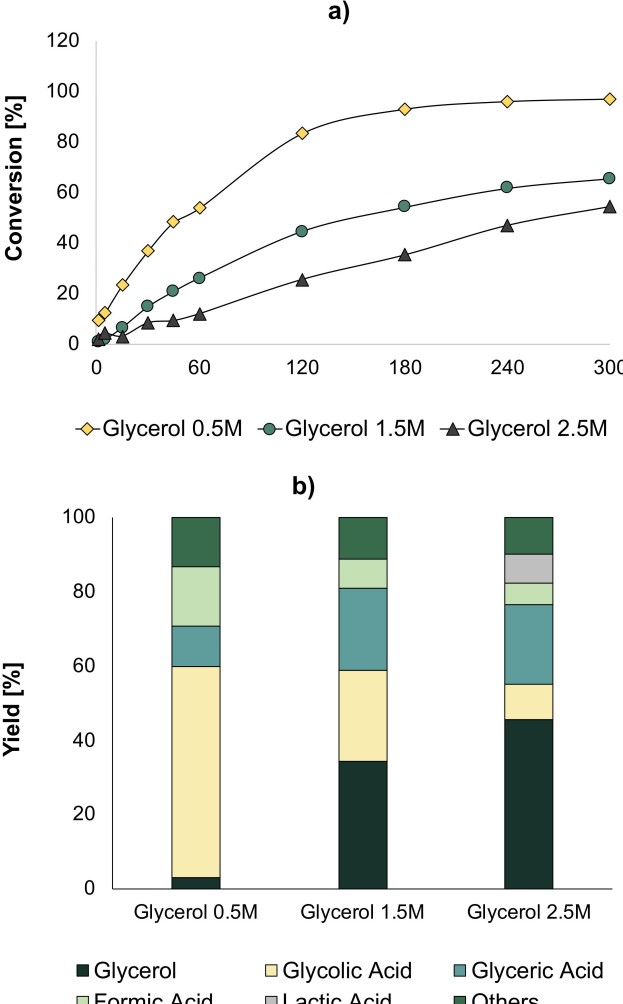

**Figure 7.** (**a**) Conversion and (**b**) products' yields obtained by increasing glycerol concentration at optimal conditions of temperature (60 °C), NaOH/glycerol ratio, and amount of catalyst (0.7 g).

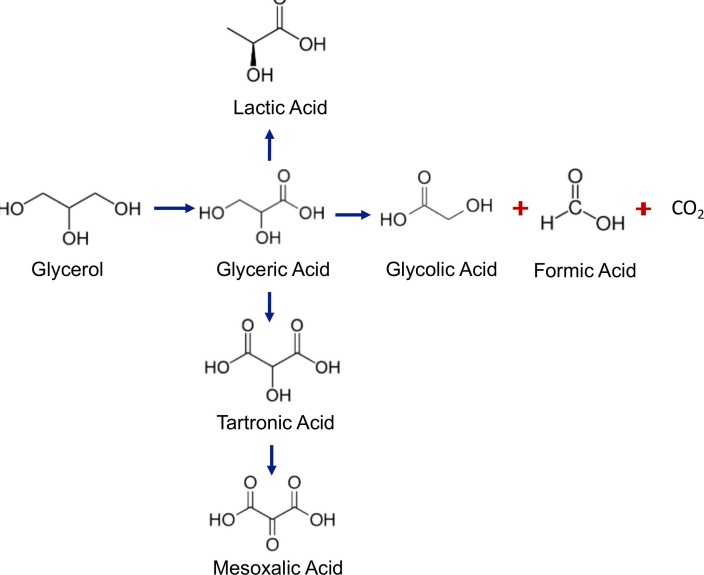

**Figure 8.** Scheme of the reaction mechanisms from the oxidation of glycerol in a basic medium. Figure produced by the present authors based on [3,14,16,32,33].

In order to increase the conversion, tests were performed with 1.5 M glycerol, while increasing the amount of catalyst. Catalyst amounts were chosen from 0.1 to 2 g. Increasing the catalyst amount led to a drastic increase in glycerol conversion, with a value of almost 100% when using 2 g of catalyst. This full conversion was linked with a change in terms of selectivity, as depicted in Figure 9b. With the lowest quantity of catalyst, the main products were FA, GLYA, and GA, while when the quantity of catalyst was larger than 0.9 g, new products appeared, such as mesoxalic acid (MA) and LA, while the quantity of $CO_2$ (total oxidation) drastically increased (Figure 8).

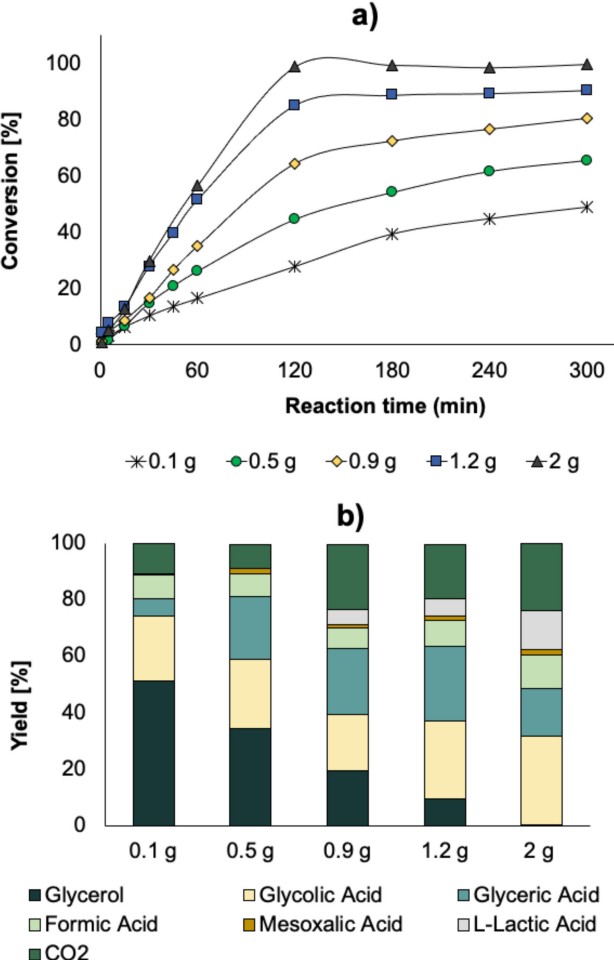

**Figure 9.** (**a**) Conversion for different amounts of catalyst as a function of time, and (**b**) product yields obtained at maximum conversion by increasing the amount of catalyst under previously determined optimal reaction conditions.

Taking into account that LA was not targeted in this reaction and also because it is quite difficult to separate it from GLYA, reducing the amount of formed LA is important. For doing this, a series of subsequent catalytic tests were performed by increasing the $O_2$ partial pressure (higher amount of available oxygen in the liquid phase). The reaction parameters were 60 °C, NaOH/Glycerol ratio = 2, and 2 g of catalyst. Figure 10 shows the results of these tests.

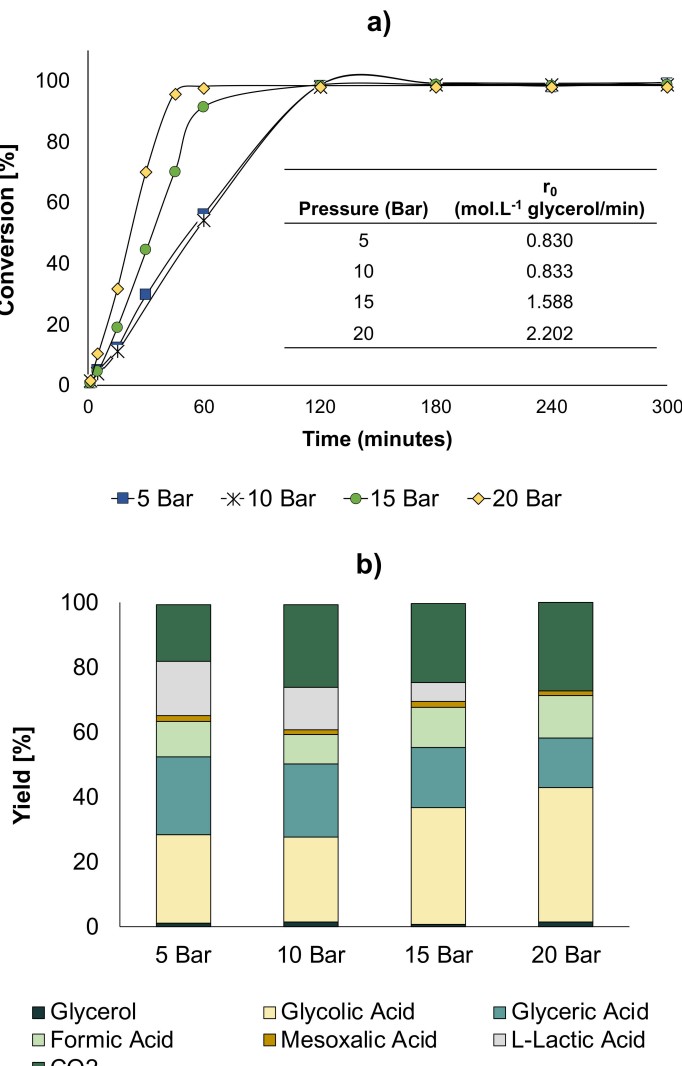

**Figure 10.** Production of glycolic acid at different $O_2$ pressures under optimal conditions determined in the previous experiments: (**a**) conversion of glycerol and initial reaction rates, and (**b**) product yields.

The results showed that while an increase in $O_2$ pressure to 10 bar did not yield any significant change in conversion and yields, at 15 bar, a higher reaction rate was actually observed together with an increase in GLYA and a decrease in LA selectivities (Figure 10a,b). When the pressure was further increased to 20 bar, the reaction rate was even higher than that observed at 15 bar. Further, at this pressure, the GLYA yield increased, while LA was not produced anymore. Furthermore, it was evidenced that as the pressure of $O_2$ increased, the initial reaction rate proportionally increased. Thus, at a pressure of 20 bar, in addition to favoring the production of GLYA, complete conversion was obtained at a shorter reaction time.

As the availability of oxygen and the amount of catalyst was identified as an important feature, we decided to carry out the evaluation on the catalytic performances of two more variables: the $O_2$/glycerol ratio and the catalyst/glycerol mass ratio. The $O_2$/glycerol ratio is indicative of the oxygen availability, and was measured in terms of pressure in the sky of the reactor. We thus carried out CCD 2. In this design, ten observations were proposed. The levels of each variable are shown in Table 6. These levels were defined according to previous experiences in our laboratory, conditions and technical limitations of the set-up, and according to bibliography. For the $O_2$ pressure/glycerol ratio, ratios from 0.01 to 0.47 were evaluated, corresponding to pressures from 1 to 40 bar. For the catalyst/glycerol ratio (*g/g*), ratios from 0.03 to 0.13 were evaluated, which corresponded to catalyst quantities from 0.9 g to 3.6 g. These tests were performed with a glycerol solution

of 1.5 M and at NaOH/glycerol molar ratio of 2 (optimal condition found in the first stage). The experimental tests results were analyzed by an ANOVA analysis, in which the most influential variables and their interaction were found.

**Table 6.** Variables considered in CCD 2.

| Variable | Minimum Level | Maximum Level |
|---|---|---|
| $O_2$ pressure/glycerol ratio (mol/mol) | 0.01 | 0.47 |
| Catalyst/glycerol ratio (g/g) | 0.03 | 0.13 |

The results obtained with each operating condition with CCD 2 can be seen in Table 7. The statistical analysis showed the influence of the two variables and the interaction between them.

**Table 7.** Experimental results obtained at each operating condition of CCD 2.

| Observation | Ratio $O_2$/Glycerol | Ratio Catalyst/Glycerol | Glycolic Acid Yield |
|---|---|---|---|
| Obs1 | 0.06 | 0.03 | 19.64 |
| Obs2 | 0.4 | 0.03 | 33.04 |
| Obs3 | 0.06 | 0.11 | 8.49 |
| Obs4 | 0.4 | 0.11 | 25.35 |
| Obs5 | 0 | 0.07 | 0 |
| Obs6 | 0.47 | 0.07 | 38.15 |
| Obs7 | 0.23 | 0.013 | 15.8 |
| Obs8 | 0.23 | 0.13 | 24.12 |
| Obs9 | 0.23 | 0.07 | 32.48 |
| Obs10 | 0.23 | 0.07 | 40.88 |

Figure 11 shows the surface response for the GLYA yield evaluated against $O_2$/glycerol molar ratio and catalyst/glycerol mass ratio. For low catalyst/glycerol and oxygen/glycerol ratios, the GLYA yield was low. This was mainly due to the fact that with less oxygen availability, the selectivity of the reaction turned to other products, such as LA. On the other hand, the maximum GLYA yield was found at intermediate values of catalyst/glycerol (0.07) and also oxygen/glycerol (0.23) ratios. This was due to the fact that when the amount of catalyst is increased, it is necessary to increase in parallel the $O_2$ pressure to have sufficient oxygen. However, due to technical limitations, the maximum $O_2$/glycerol ratio studied herein did not allow using sufficient oxygen pressure for the maximum amount of catalyst, and it was not possible to increase the pressure even further. Due to this relative deficiency of oxygen at high catalyst amounts, undesirable products were obtained that affected the GLYA yield.

The optimal values we found were $O_2$/glycerol ratio of 0.23 and catalyst/glycerol ratio of 0.07.

Once the optimal conditions of temperature (60 °C), NaOH/glycerol (2), $O_2$/glycerol ratio (0.23), and catalyst/glycerol ratio (0.07) were determined, the glycerol concentration was increased, and we varied it from 1.5 to 3 M.

The results showed a significant improvement in the conversion. If we compare the effects of the conditions, for a glycerol concentration of 1.5 M without optimal conditions, a conversion of 60% was obtained at 300 min (Figure 7a), while with the optimal conditions, a conversion of almost 100% after 45 min was obtained (Figure 12a). Comparing the concentration of glycolic acid obtained with an initial solution of 0.3 M glycerol (0.27 mol/L), it was observed that by increasing the initial concentration of glycerol under optimal conditions, a higher concentration of glycolic acid was obtained—0.93 mol/L when a 1.5 M solution was used and 0.88 mol/L when using a 2 M solution.

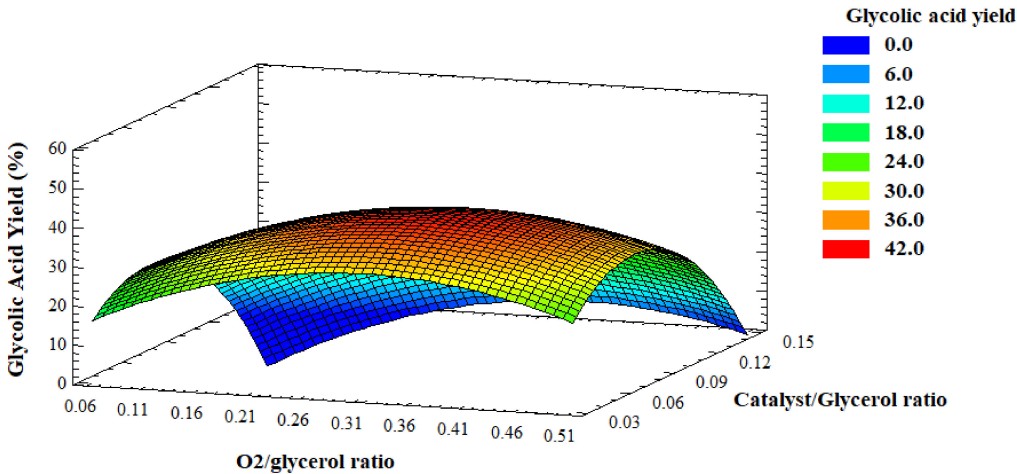

**Figure 11.** Surface response for the glycolic acid yield evaluated against the most influential variables, namely the $O_2$/glycerol molar ratio and the catalyst/glycerol mass ratio.

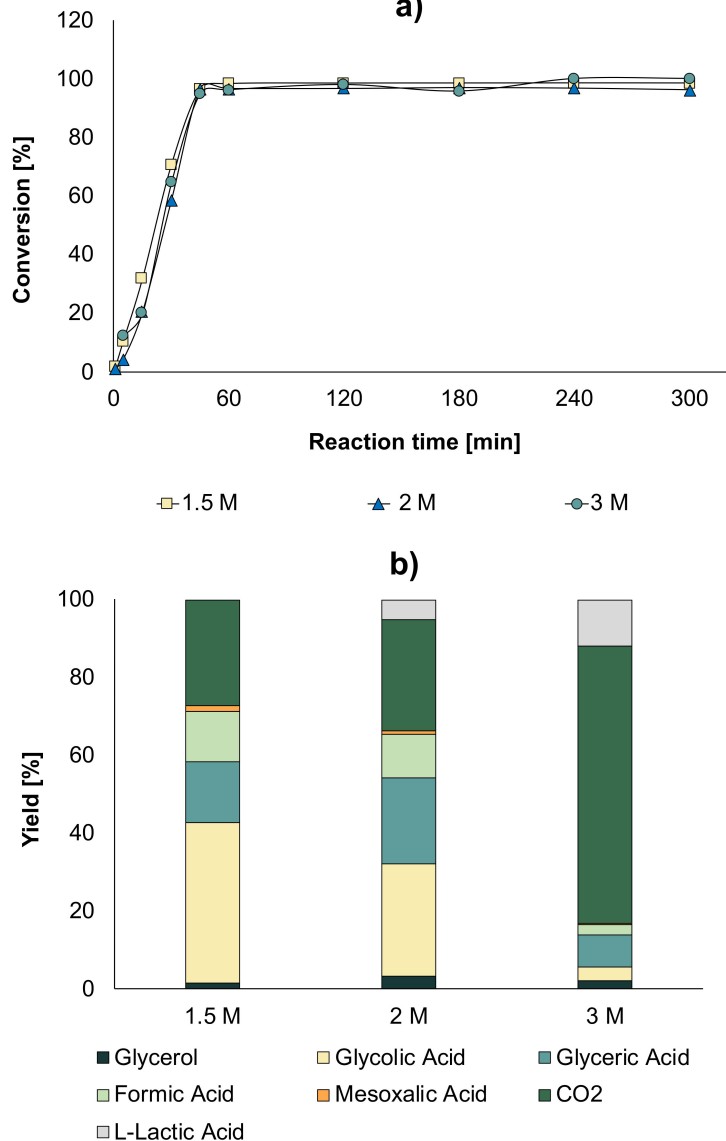

**Figure 12.** (**a**) Conversion of glycerol as a function of time for different glycerol concentrations, (**b**) product yields at maximum conversion by increasing the glycerol concentration under optimal conditions of temperature (60 °C), NaOH/glycerol (2), $O_2$/glycerol ratio (0.23), and catalyst/glycerol ratio (0.07).

The distribution of products in the liquid at maximum conversion is given in Figure 12b. As the glycerol concentration increased, MA appeared as a product that was not observed when a glycerol concentration of 0.3 M (Figure 5b) and 0.5 M were used (Figure 7b). In addition, it was found that, as the concentration of glycerol increased, LA appeared in a higher concentration, which was an undesired product in our study.

The quantity (mol) of GLYA produced per liter of solution was evaluated at different concentrations of glycerol under optimal conditions and compared with the initial conditions (Figure 13). A significant increase in the concentration of GLYA was observed when using a 1.5 M and 2 M solution, leading to the possibility of increasing the glycerol concentration and obtaining a solution rich in the product of interest.

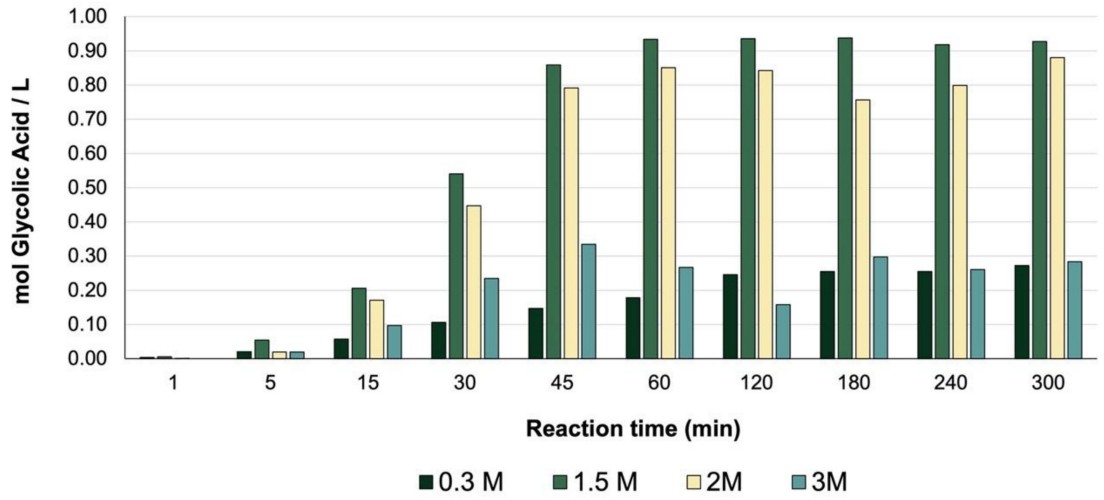

**Figure 13.** Concentration in the liquid product obtained by increasing the glycerol concentration at optimal operating conditions of temperature (60 °C), NaOH/glycerol (2), $O_2$/glycerol ratio (0.23), and catalyst/glycerol ratio (0.07).

## 3. Materials and Methods

### 3.1. Catalyst Synthesis and Characterization

The catalyst used in the experimental tests was a silver catalyst supported on a mixed cerium-zirconium oxide with the $Ce_{0.75}Zr_{0.25}O_2$ composition.

As a first step, the $Ce_{0.75}Zr_{0.25}O_2$ support was synthesized by a sol-gel method reported by Rossignol et al. [34]. Cerium nitrate $Ce(NO_3) \cdot 6H_2O$ (Sigma Aldrich, 99%) was used as the cerium precursor, and a solution of zirconium *n*-peroxide $Zr(OC_3H_7)_4$ in 1-propanol (Sigma Aldrich) was used as the zirconium precursor. The necessary amounts of each precursor were calculated to maintain a Ce-Zr ratio of 3:1.

Cerium nitrate was dissolved in water at room temperature and under stirring, while the zirconium *n*-peroxide solution was diluted in isopropanol. The aqueous cerium nitrate solution was added to the *n*-peroxide zirconium solution under stirring, with consequence of immediately obtaining the pseudo-gel.

Subsequently, the gel was dried at 60 °C for one hour, and then at 110 °C overnight. Finally, the powder obtained was calcined under dynamic air atmosphere at 550 °C for 4 h (5 °C/min).

Once the support was obtained, we proceeded to silver deposition, using the deposition–precipitation method [21,35]. Methanol (Sigma Aldrich, 99%) was added to the support placed in a round-bottom flask. This solution was then placed under reflux at the temperature of 65 °C for one hour. Silver nitrate $AgNO_3$ (Sigma Aldrich, 99%) was used as the silver precursor. The amount of silver nitrate necessary to obtain a silver loading of 5 wt.% was diluted in methanol and added dropwise to the support solution. Then, a commercial 36% formaldehyde solution in $H_2O$ (Sigma Aldrich, 36%) was added so as to obtain a formaldehyde/silver molar ratio of 10. Once the formaldehyde was added, the solution turned dark brown. This solution was left under reflux at the same temperature

(65 °C) under stirring for one more hour. The pH of the solution was then adjusted to 8 by adding a 0.3 M solution of sodium hydroxide NaOH (Sigma Aldrich, 98%); reflux and stirring were then kept for an additional hour. Finally, the solution was cooled to room temperature and filtered. The recovered product was washed with ultrapure water and dried at 110 °C for 24 h, thus obtaining the catalyst with 5 wt.% of Ag on $Ce_{0.75}Zr_{0.25}O_2$ support ($5\%Ag/Ce_{0.75}Zr_{0.25}O_2$).

The synthesized catalyst was characterized by various techniques. Its textural properties were evaluated through nitrogen physisorption ($-196$ °C) using the BJH method with a Micromeritics TriStar II 3020.

X-ray diffraction was used to check the sample crystallinity. The analysis was performed on a Bruker AXS D8 advanced diffractometer with $CuK\alpha$ source ($\lambda = 0.154$ nm), in a $2\theta$ range from 20° to 90° with a step of 0.02° and scan time of 0.5 s per step. The diffractogram was analyzed using the DifracEva software and its database, comparing it with the files corresponding to the metallic silver and the CeZrO support. The average crystallite size of the synthesized support was determined using the Sherrer equation [24].

Likewise, to check the oxidation state of silver and the catalyst surface composition, X-ray photoelectron spectroscopy (XPS) was performed, using an AXIS Ultra DLD Kratos spectrometer equipped with a monochromatic aluminum source ($AlK\alpha = 1486.6$ eV). All the binding energies were calibrated with C1s being fixed at 285.0 eV. The results were analyzed and treated using the CasaXPS software.

### 3.2. Catalytic Tests

The catalytic tests were performed in the test rig shown in Figure 14. This setup consisted of a batch reactor, a mechanical stirrer, and an oxygen pressure tank, as well as temperature and pressure controls. The oxygen from the cylinder was fed into the reservoir, in which oxygen was stored and thermoregulated at 40 °C. The batch reactor was made of stainless steel and had a volume of 300 mL. The reactor temperature was controlled by a temperature control system. In addition, the reactor had an inlet for oxygen connected to the reservoir and two outlets—one for sampling and the other for depressurizing the system once the reaction is complete.

For the oxidation reaction of glycerol in the liquid phase, 200 mL of a glycerol-water solution was loaded in the reactor. The concentration of this solution was adjusted according to the desired experimental conditions. To this solution, sodium hydroxide was added with the desired NaOH/glycerol molar ratio. The reactor was closed and heated at the required reaction temperature. Once this temperature was reached, the catalyst was added with the desired glycerol/catalyst mass ratio. At this stage, the agitation started, and the system was pressurized with $O_2$; this time corresponds to the $t_0$ of the reaction.

Through the sampling valve, samples were taken at different times, during three hours of reaction. The samples were then acidified with a 2.4 M sulfuric acid solution, and subsequently analyzed on an Agilent 1260 HPLC equipped with a RID detector. For this analysis, a Rezex ROA-Organic Acid $H^+$ column (Phenomenex, 8%, $300 \times 7.8$ mm) was used at a temperature of 35 °C. A 2.5 mM solution of sulfuric acid in ultrapure water was used as the mobile phase applying a flow of 0.5 mL/min.

To identify the retention times, high purity standards of the possible reaction products were injected. For quantification, a calibration curve was performed with each product. For this, high purity standards were used, and solutions were prepared at different concentrations. The response factors were calculated for each product from the calibration curves.

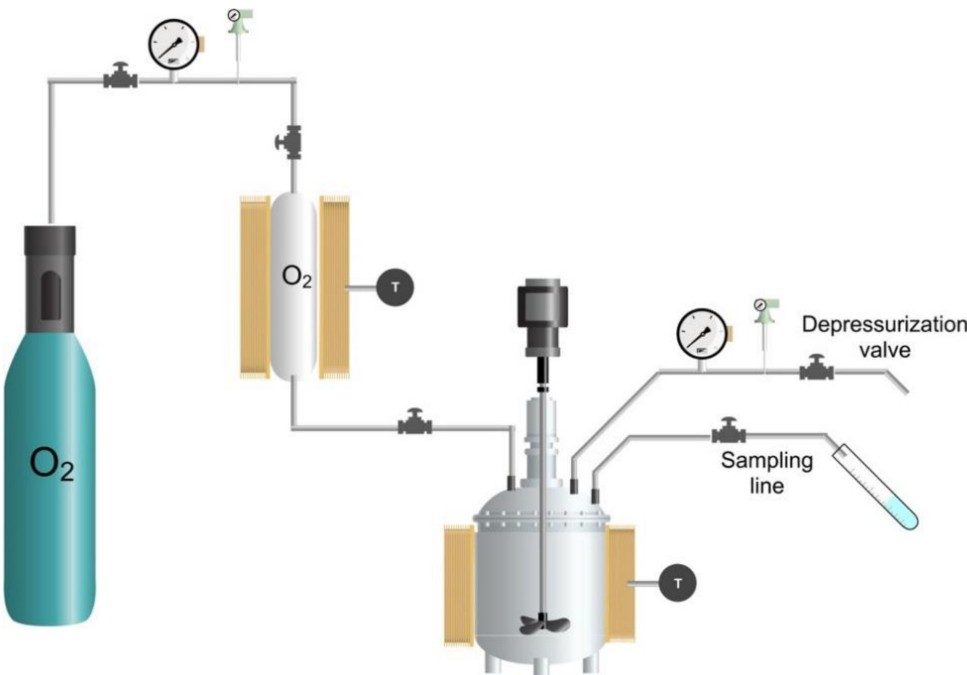

**Figure 14.** Experimental set-up used for catalytic tests.

## 4. Conclusions

To study the behavior of the catalyst and compare it with those studied in other investigations, a preliminary benchmark test was carried out under the same initial conditions evaluated in these investigations. After two hours of reaction, we observed almost full conversion (>98%) with a high selectivity to GLYA (69%). The conversion obtained in this study was higher than those obtained in other studies under the same conditions and different catalysts, concluding that the catalyst 5% $Ag/Ce_{0.75}Zr_{0.25}O_2$, in addition to presenting a complete conversion, presents a high selectivity towards GLYA [3,13], leading to GLYA yield of almost 68%.

It was then found that catalytic production of glycolic acid (GLYA) from glycerol oxidation using silver deposited on cerium zirconium oxide as a catalyst appeared to be a promising route of valorization of glycerol. In order to take this process to an industrial scale, the optimization of operating conditions was carried out. For this, some variables were studied, the most influential being the glycolic acid yield, and the optimal conditions of each of these variables were found.

It could be concluded that the production of GLYA depends mainly on glycerol concentration, NaOH/glycerol ratio, catalyst/glycerol ratio, and $O_2$/glycerol ratio. The optimal conditions we found with our setup and its limitations were: temperature 60 °C, NaOH/glycerol molar ratio of 2, $O_2$/glycerol molar ratio of 0.23, and catalyst/glycerol mass ratio of 0.07. The GLY concentration was increased, varying from 0.3 M to 3 M. It could be concluded that at the optimal conditions found, the glycerol concentration can be increased from 0.3 M to 2 M with a conversion of 100%. It was found that at optimal conditions, if the glycerol concentration is increased, the production of $CO_2$ increases proportionally. However, when reviewing the concentration of GLYA in the liquid product obtained, a significant increase in GLYA concentration was found when the glycerol concentration was increased to 1.5 M (0.93 mol GLY/L) and 2.0 M (0.88 mol GLY/L), obtaining a liquid fraction rich in GLYA.

**Author Contributions:** Writing—original draft preparation, C.P.T.R.; writing—review and editing, C.P.T.R., F.D., M.C.; supervision, F.D., M.C.; project administration, M.C.; funding acquisition, M.C. All authors have read and agreed to the published version of the manuscript.

**Funding:** This research was funded by I Site U Lille.

**Institutional Review Board Statement:** Not applicable.

**Informed Consent Statement:** Not applicable.

**Acknowledgments:** Authors acknowledge the support from Chevreul Institute (FR 2638), Ministère de l'Enseignement Supérieur, de la Recherche, et de l'Innovation, Région Hauts-de-France and FEDER. Authors want to thank O. Gardoll, P. Simon and M. Trentesaux for the technical supports. We also want to thank to Department of Investigations of Universidad de Investigación y Desarrollo.

**Conflicts of Interest:** The authors declare no conflict of interest.

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
