# Peer review of "Catalytic Production of Glycolic Acid from Glycerol Oxidation: An Optimization Using Response Surface Methodology†"

_catalysts, doi:10.3390/catal11020257_

Round 1
Reviewer 1 Report
Article quite interesting for Catalyst readers, in the field of catalysis, for the glycerol oxidation, a major-value in the glycerol valorisation sustainable tool.
Nevertheless, it's necessary to improve the article in the follow aspects:
1) Correct some english gramamr and sentences;
2) Introduction and conclusions sections can be slightly improved, with mpre references to support them;
3) At page 13, figure 9, it's missing any references to support authors conclusions regarding mechanism aspects, like, the secondary reactions occurence. It's not enough self considerations of the authors;
4) Figures 7 and 12 can be presented with more quality (better resolution images).
Author Response
Response to the Reviewer’s comments and suggestions
We would like to thank the reviewers for taking the time to review and carefully check our manuscript.
The minor corrections requested by the 2 reviewer have been made; note that the reference numbers and page numbers in this letter refer to revised version of the manuscript. All changes are presented in red in the manuscript.
Reviewer #1
Article quite interesting for Catalyst readers, in the field of catalysis, for the glycerol oxidation, a major-value in the glycerol valorization sustainable tool. Nevertheless, it's necessary to improve the article in the follow aspects:
- Correct some English grammar and sentences
ANSWER: We thank the reviewer for the suggestion, and we take it with pleasure. We have double-checked the paper and corrected all grammar errors.
- Introduction and conclusions sections can be slightly improved, with more references to support them.
ANSWER: We appreciate the comment; based on this, we revised and expanded the introduction by adding information of research studies about different routes of glycerol valorization.
The conclusions were revised and improved, expressing in a clearer way the most relevant results of the investigation. Changes are highlighted in red within the document.
- At page 13, figure 9, it's missing any references to support authors conclusions regarding mechanism aspects, like, the secondary reactions occurrence. It's not enough self-considerations of the authors;
ANSWER: We thank the reviewer for the comment and proposed improvement regarding this figure. We had indeed omitted the references, for which we apologize. References were added at the bottom of the figure.
- Figures 7 and 12 can be presented with more quality (better resolution images).
ANSWER: We thank the reviewer for this comment. Considering that the software in which the figure was made does not offer higher resolution, we decided to redraw the figures using another software. Accordingly, the quality of Figures 7 and 12 was considerably improved.
Reviewer 2 Report
Capron et al. present and discuss their work on the Ag/Ce oxide-catalysed oxidation of glycerol to glycolic acid in a batch reactor system. Besides catalyst synthesis and characterization, the authors also provide insights into the theoretical and experimental optimization of the reaction conditions to achieve improved GLYA production. Substrate, additive and catalyst dosage was varied as well as process temperature.
The English language of the manuscript is good and does not need to be improved. The scientific content, both analytical and experimental, is well balanced and gives good insights into the glycerol oxidation. I can recommend the manuscript as it is in its current version. Possibly as one comment: in the introduction chapter, the authors might give some information about the use of glycerol as hydrogen source via its oxidation to dihydroxy acetone and the role of the latter one in the cosmetic industry.
Author Response
Response to the Reviewer’s comments and suggestions
We would like to thank the reviewers for taking the time to review and carefully check our manuscript.
The minor corrections requested by the 2 reviewer have been made; note that the reference numbers and page numbers in this letter refer to revised version of the manuscript. All changes are presented in red in the manuscript.
Reviewer #2
Capron et al. present and discuss their work on the Ag/Ce oxide-catalysed oxidation of glycerol to glycolic acid in a batch reactor system. Besides catalyst synthesis and characterization, the authors also provide insights into the theoretical and experimental optimization of the reaction conditions to achieve improved GLYA production. Substrate, additive and catalyst dosage was varied as well as process temperature.
The English language of the manuscript is good and does not need to be improved. The scientific content, both analytical and experimental, is well balanced and gives good insights into the glycerol oxidation. I can recommend the manuscript as it is in its current version. Possibly as one comment: in the introduction chapter, the authors might give some information about the use of glycerol as hydrogen source via its oxidation to dihydroxy acetone and the role of the latter one in the cosmetic industry.
ANSWER: We thank the reviewer for such positive comments, and we are glad that our research work met your appreciation. We have addressed your comments; we thus added information about different routes of glycerol valorization. Changes are highlighted in red within the document.